# An Exploratory Study of Nutrition Knowledge and Challenges Faced by Informal Carers of Community-Dwelling People with Dementia: Online Survey and Thematic Analysis

**DOI:** 10.3390/geriatrics8040077

**Published:** 2023-07-22

**Authors:** Abdeljalil Lahiouel, Jane Kellett, Stephen Isbel, Nathan M. D’Cunha

**Affiliations:** 1School of Rehabilitation and Exercise Sciences, Faculty of Health, University of Canberra, Bruce, ACT 2617, Australia; u3148325@uni.canberra.edu.au (A.L.); jane.kellett@canberra.edu.au (J.K.); stephen.isbel@canberra.edu.au (S.I.); 2Ageing Research Group, Faculty of Health, University of Canberra, Bruce, ACT 2617, Australia

**Keywords:** nutrition knowledge, informal carers, dementia, nutrition education, qualitative research, questionnaire

## Abstract

Nutrition knowledge is a primary factor influencing food choices and the ability to identify nutritional risk for carers of people with dementia. Acquiring nutrition knowledge helps carers monitor changes in food intake and micronutrient intake, and whether a healthy and balanced diet is being consumed. This study aimed to assess the nutrition knowledge of carers in the Australian community and their experiences with nutrition education. Using a mixed-methods approach, the nutrition knowledge of informal carers was assessed using the revised General Nutrition Knowledge Questionnaire (AUS-R-NKQ), and interviews of informal carers were used to explore the perspectives in a sub-sample. A total of 57 carers (44 females; mean age of 63.0 ± 13.1) completed the survey, and 11 carers participated in follow-up interviews. The total sample scored 22.9 (±4.57) out of 38 on the AUS-R-NKQ, suggesting basic nutrition knowledge. The interviewed carers acknowledged the importance of healthy nutrition but viewed the provision of a healthy diet for a person with dementia as challenging. In both the survey and interviews, carers reported limited use and availability of dementia-specific nutrition resources. Carers were unsatisfied with the advice and number of referrals provided to improve the nutrition of the person with dementia and desired less confusing nutrition education materials adapted to their level of knowledge. The present study highlights the need for informal carers to be supported to acquire adequate nutrition knowledge.

## 1. Introduction

Dementia is a clinical syndrome characterised by progressive deterioration in cognitive functioning that interferes with daily life [1]. Dementia is the second leading cause of death and third leading cause of burden of disease in all Australians and the leading cause of death and burden of disease among women [2]. Risk factors include diabetes, hypertension, obesity, oxidative stress, depression, head trauma, smoking, physical inactivity, poor nutrition and diet [3]. Currently, there is no cure for dementia, and symptomatic treatments, such as cholinesterase inhibitors, are commonly prescribed [4]. Two types of care coexist: formal care, referring to paid care provided in healthcare settings such as residential aged care or by health professionals; and informal care, provided by unpaid family members or relatives [5]. Globally, informal care plays an important role in supporting people with dementia in their daily activities [6]. Estimates have suggested that informal carers of people with dementia living in the community equal 40 million full-time workers, which is predicted to increase to 65 million by 2030 [7]. Without informal carers, community-dwelling people with dementia would have poorer quality of life and increased hospitalisation rates, resulting in a significant economic burden [8]. However, few informal carers receive appropriate training and support, increasing their risk of physical injuries, heightened stress, and emotional exhaustion [9].

The absence of an effective treatment has resulted in further attention being paid to lifestyle modifications, including a healthy diet, and physical and cognitive activity, which may contribute to slowing the rate of cognitive decline [10]. Nutrition in people with dementia is an important facet of dementia care, and eating disturbances such as physiological and psychological changes, nutritional difficulties, and weight loss can affect eating behaviours and nutritional status [10,11,12]. Moreover, although cognitive impairment results from the interaction of multiple factors, poor nutritional status is associated with higher risk of cognitive decline [13,14,15], and nutritional deficiencies may accelerate oxidative damage in the brain, contributing to more rapid cognitive decline [16]. During the early stages of dementia, eating behaviours often initially change due to memory problems, which can contribute to issues in preparing foods or remembering to eat and stay hydrated [10]. Sensory impairments can also contribute to a change in eating behaviours due to loss of smell or taste. In addition, changes in behaviour and functional capacity contribute to malnutrition risk due to depression, social isolation, and loneliness [10]. As dementia progresses, physiological changes such as dysphagia can further contribute to a reduction in energy intake and weight loss, further exacerbating cognitive decline [17].

Feeding difficulties experienced by the person with dementia increase carer burden and anxiety, affecting the quality of care provided [8,14]. However, carers of people with dementia lack adequate nutrition training and support to identify malnutrition risk and address nutritional needs [18,19]. There is limited research on the perceptions and experiences of informal carers regarding nutrition-related issues and the strategies they use. Existing research has shown that informal carers are provided with inadequate information and support from healthcare professionals regarding nutrition-related issues such as feeding, weight loss identification or the ability to encourage appetite in people with dementia [12,20]. In severe dementia, healthcare professionals should consult informal carers on nutrition and hydration in the decision-making process to act in the best interest of the person with dementia [21]. Furthermore, nutrition knowledge contributes to mitigating carer burden by reducing stress and preventing physical and mental health decline in carers [22]. Nutrition knowledge is a significant factor influencing food choices, affecting the ability to identify nutritional risk, and aids informal carers in monitoring whether a healthy diet is being consumed [19].

To our knowledge, no studies have attempted to assess carers’ nutrition knowledge in the Australian community or describe their preferred methods for acquiring nutrition knowledge and the potential support they need. For carers of people with dementia to provide optimal nutrition, it is important that they are equipped with appropriate knowledge about diet quality, maintaining energy intake, identifying nutritional risks, and strategies to promote the consumption of healthy foods, to maintain the overall well-being and health of the person they care for [14,23]. Therefore, the aim of this study was to provide insights into the nutrition knowledge of informal carers of people with dementia, common nutrition-related issues, and preferred delivery modes of nutrition education to potentially reduce the risk of nutritional deficiencies and nutrition-related issues, such as malnutrition in the person with dementia. The research questions are the following:What is the nutrition knowledge of informal carers of people with dementia?What are common nutrition-related issues faced by informal carers of people with dementia?What are the preferred modes of delivery of nutrition education for informal carers of people with dementia?

This research has the potential to inform the design of interventions to improve the nutrition knowledge of carers of people with dementia, which can reduce the risk of malnutrition in the dyad.

## 2. Materials and Methods

### 2.1. Study Design

This mixed-methods study was informed by a two-phase sequential explanatory design [24] and was conducted between November 2022 and April 2023. The initial quantitative phase of data collection was a cross-sectional online survey to characterise a convenience sample of carers of people living with dementia and their level of nutrition knowledge. The survey was hosted and distributed using Qualtrics software (Qualtrics LLC, Provo, UT, USA). The survey was designed to assess informal carers’ nutrition knowledge, understand the type of issues they are facing, the type of nutrition support they are providing, and their preferred methods for receiving nutrition-related knowledge. Based on piloting with older people, the survey takes approximately 25–30 min. In the qualitative phase, we completed semi-structured interviews of informal carers of people with dementia living in Australia to gain deeper insights into the challenges faced by this population.

### 2.2. Participant Selection

Potential participants were of any gender, aged 18 years and over, and currently providing informal dementia care to a community-dwelling person with dementia living in Australia. Formal carers of people with dementia, providing professional, paid care were excluded. To recruit participants, we primarily contacted facilitators of carer support groups across Australia via email, phone, or mail to request assistance in distributing flyers. The study was advertised on the Dementia Australia and StepUp for Dementia websites and across social media. The survey was also advertised in carer-related newsletters, snowballing, and word of mouth. All participants provided informed consent before commencing the survey.

### 2.3. Data Collection

#### 2.3.1. Quantitative Data

The 77-item survey consisted of open-ended questions, matrix-table questions, multiple-choice questions, slider questions, form-field questions, and yes/no questions across four sections:(1)Sociodemographic information.(2)Nutrition knowledge was assessed using the revised General Nutrition Knowledge Questionnaire for Australia (AUS-R-NKQ). The AUS-R-NKQ was redesigned and validated in 2020 after advances in understanding the diet–disease relationship and changes in nutrition recommendations [25]. The General Nutrition Knowledge Questionnaire (GNKQ) is validated internationally and in Australia [26,27,28]. The thirty-eight-item questionnaire consists of four categories:Dietary recommendations: Eleven questions focussed on the Australian Dietary Guidelines.Nutrients in foods: Nine questions relating to the different food groups (meats, vegetables, fruits, grains, and dairy products) and the nutrients they contain (carbohydrates, proteins, fats, fibre, minerals, and vitamins).Food choices: Nine questions assessing knowledge of healthy food choices.Diet–disease relationships: Nine questions about the interaction between diet and nutrition, and common health problems and diseases.

Higher scores indicate greater nutrition knowledge [25].

(3)Nutritional care and provision and nutrition-related issues in people with dementia.(4)Preferred delivery modes of nutrition education programmes for informal carers.

#### 2.3.2. Qualitative Data

In the survey, respondents were asked if they were interested in participating in an additional semi-structured interview on Zoom. Using a random number generator, 18 carers were invited to a semi-structured audio-recorded interview at a time convenient to the participant. The random sample method was used to reduce the perceived burden of participation, as carers were only invited to complete an interview by chance. Six open-ended questions were posed to participants, informed by survey results and a review of the published literature [12,14,29], to explore their perceptions regarding nutrition education and nutrition challenges being faced, their impressions about support from healthcare professionals, the type of support they wanted and needed, and their experiences with families and peers (Figure 1). Qualitative data are reported using the Consolidated Criteria for Reporting Qualitative Research Checklist (COREQ) [30].

### 2.4. Data Analysis

To achieve a representative sample size of carers, 384 participants (95% confidence interval and 5% margin of error) were required. However, as this was an exploratory study, a convenience sample was used after attempting to contact all potential carer support groups to advertise the study.

Survey data were checked for completeness and then exported to IBM SPSS Statistics for Windows (version 29.0). Descriptive statistics were used to present a summary of quantitative data. Categorical variables are provided as frequencies. Numerical variables, such as age and AUS-R-NKQ scores, are presented as means with standard deviation. After testing the normality of the data, independent t-tests were used to report differences between females and males, and for categorical variables, we used the chi-square test of independence.

The qualitative analysis was conducted by a four-person multidisciplinary team (three males; one female). The primary interviewer was an Honours student (A.L.) with qualifications in public health and dentistry and three months of experience assisting in a study for people with dementia as an undergraduate student. Before the first interview, an experienced author with qualifications in human nutrition, and geriatrics and gerontology assessed a mock interview (N.M.D). After refinement and feedback, interviews with carers were scheduled and completed. All participants were naïve to the researchers. Interviews were recorded and transcribed verbatim before analysis. A reflexive thematic analysis was performed using the six-phase guide by Braun and Clarke [31]. This involved an inductive, iterative process of data reduction and systematic comparison to generate themes or concepts from the text provided [32,33]. Text was coded independently by two researchers (A.L and N.M.D) into categories and further broken down into sub-categories in accordance with the depth of responses provided using Microsoft Word. Data collection continued until data saturation was reached [34]. After reviewing categories, themes were generated based on the interpretation of the data and named to produce the results. As the themes were developed, the reasons, assumptions, and process of developing them were discussed with other experienced qualitative researchers with qualifications in dietetics (J.K.) and occupational therapy (S.I.). This was a form of reflexivity used in data analysis to strengthen the trustworthiness of the analysis.

## 3. Results

Participant characteristics are displayed in Table 1. The overall number of people to commence the survey was 104. In total, 63 (60.6%) participants provided consent and proceeded to the survey questions. The final sample included 57 (54.8%) completed questionnaires, and the mean age of participants was 63.0 (±13.1). The sample included 44 females (mean age 61.5 ± 12.1) and 13 males (mean age 68.2 ± 15.3). Survey participants lived in urban areas (68.4%); New South Wales (52.6%) was the most represented state. A high level of education was declared by 68.7% of carers, with 42.4% possessing a bachelor’s degree, and 26.3%, a higher university degree. Spouses or partners represented 45.6% of all carers, and most carers (77.2%) had provided care for at least two years. While most male carers were retired (76.9%), most female carers were still working (59.1%). Only one carer had previously studied nutrition, and most carers (73.7%) were responsible for the household’s grocery shopping.

Selected survey results are presented in Table 2. In total, 63.2% of the sample reported providing a healthy diet to the person they cared for, and this was slightly higher in females (63.6%) than males (61.5%). The majority (78.8%) of carers experienced at least one nutrition-related issue, and 87.7% of carers accessed a nutrition resource.

Self-managed education programmes were the preferred method of acquiring nutrition knowledge for this sample (36.8%), followed by online one-on-one tutoring (12.3%) and telehealth consultations (12.3%) (Figure 2).

The average score of the sample on the AUS-R GNKQ was 22.9 (±4.57) or 60.2% with a range of 11–34, and a median of 23 out of 38. The highest-scoring sections were knowledge of food choices, with 7.94 (±0.833) out of 9, and the Australian Dietary Guideline section, with 8.87 (±2.06) out of 11. However, participants had more difficulty with the food groups and their nutrients, and the diet–disease relationship sections, scoring 4.33 (±1.91) out of 9, and 4.52 (±1.33) out of 9, respectively. Overall, females performed better than their male counterparts (*p* = 0.007), averaging 23.8 (±4.51) or 62.6% compared with 20.0 ± 3.54 or 52.6%, respectively. Since diagnosis, carers observed an increased consumption of sugary foods (47.3%), decreased meat and animal protein consumption (50.8%), and no changes in salty foods consumption (66.6%). Weight loss since diagnosis was reported by 54.3% of carers, and most carers were facing at least one nutrition-related issue (78.9%).

### 3.1. Interviewed Participants

In total, 18 carers of community-dwelling people with dementia were invited to participate in interviews, and 11 of them agreed to participate. Of those who did not participate, one accepted but later declined; one initially accepted but was subsequently not contactable; and five carers did not respond to the invitation.

The interviews varied between 15 and 44 min, with an average duration of 21:55 min. The average age of participants was 64.8 (±8.92) years, and carers were spouses (n = 5), children (n = 3), family members (n = 2), and a live-in carer (n = 1). The length of time as a carer was 5–10 years (n = 5), 2–5 years (n = 5), and 1–2 years (n = 1). All eleven participants were female. The interview participants’ score on the AUS-R-NKQ was 24.0 (±3.68) on average. Their preferred methods for acquiring nutrition knowledge were self-managed education programmes (n = 6), online group sessions (n = 2), online one-on-one tutoring (n = 2), and engaging with peers during nutrition education sessions (n = 1). The characteristics of the interviewed carers are presented in Table 3.

### 3.2. Thematic Analysis

Quotes from interview participants support the identified themes from the dataset. Supporting quotations are coded as participant numbers (P1, P2, etc.). There were four prominent themes from the interviews. Table 4 provides an overview of the themes and subthemes.

#### 3.2.1. Theme 1: Providing a Healthy Diet Is Challenging for Carers

Nine participants viewed healthy-diet provision for a person with dementia as challenging due to multiple factors, such as physiological and cognitive changes in the person with dementia, prompting carers to be proactive and try to provide multiple healthy food options.

“I think his communication and ability to say what he wants and what he needs, might diminish, which means that you, you really have to kind of be looking after the food side of things and encouraging and, you know, presenting good options, rather than just waiting for them to choose something.”(P10)

##### Sub-Theme 1: Physiological Changes Make it Difficult to Provide a Healthy Diet

For nine carers, physiological changes occurring during dementia progression, such as dysphagia, anorexia, and ageusia, affected food intake. Four carers discussed poor appetite, weight loss, and loss of interest in food. These, coupled with physiological changes, can make meals unpleasant for the person with dementia.

“You, you’ve got no weight gain and loss of appetite and pressure on your gut and all that sort of stuff. So, all of those things compound to make the fact that eating probably isn’t much fun.”(P1)

Two carers reported a severe gustatory decline and that the person they care for used salt even prior to tasting food.

“He’s not tasting anything and getting the salt pot and just chucking it on the plate. So, I don’t cook with salt, or barely because I know he’s not going to taste it and chuck the salt on top of it.”(P2)

##### Sub-Theme 2: Memory Loss Contributes to Lower Energy Intake

Seven carers mentioned memory loss as part of overall energy intake difficulties. Six carers understood that a person with dementia experiencing short-term memory loss and decision-making problems has more difficulty recognising food in front of them or forgets to eat. To improve the mealtime experience, carers often prepared comforting and familiar foods.

“Sometimes, he forgets to eat when he is on his own. Sometimes I cook things he likes, something that when he was young, in his younger years, like homemade pie. He likes that because that reminds him of his young days of his childhood.”(P6)

“Or she doesn’t drink enough water we have to push her to drink water. She’s forgotten a little bit. Sometimes she forgets how to swallow.”(P5)

##### Sub-Theme 3: Financial Barriers to Healthy Nutrition

Four carers believed that the cost of fresh produce has risen after natural disasters (bushfires, droughts, and floods) and the COVID-19 global pandemic, preventing them from offering a nutritious diet.

“I mean, there have been, you know, difficulties as you know, last few years with, you know, floods and COVID and that kind of thing, trying to access fruit and veg that, you know, isn’t too expensive.”(P9)

##### Sub-Theme 4: The Provision of a Healthy Diet Accentuates Carer Burden

Six carers experienced a relatively heavy burden due to nutritional care. They had limited time to prepare nutritious food. Conciliating nutritional care and assistance with activities of daily living with their work commitments and social life resulted in increased stress. Carers emphasised the need to be supported.

“I think it’s the carers that need to be supported so they can continue doing what they’re doing. It’s tiring. People should be giving me information and helping me with what I need. I’m helping my dad; I’m helping my family—rather than me having to look out for everything. Plus trying to work full time. It’s hard.”(P5)

“Realistically, there are so many battles to be had with this condition and there’s so many things I have to think about every day. Just basic things, and it’s just one more thing to have to think about [nutrition]. […] If somebody would tell me that there was a HelloFresh [meal delivery service] for dementia, like a special diet that I could pay for, I would pay any money for that.”(P8)

Moreover, as people with dementia usually have other comorbidities, medical appointments consume considerable time. One carer reported over 50 appointments in a year, impacting the time needed to address diet quality and contributing to opting for unhealthy options.

“Making things up at times gets a little bit difficult, putting the time in. We had over 50 medical appointments last year. I was too tired last night to get any dinner last night, so we bought a hamburger and we had that with, I’d cooked corn on the cob.” (P9)

A carer felt upset because her loved one did not want to eat healthy foods when she took the time to cook for her.

“I make all of her evening meals and I make things I know that she likes, and I bulk make them and I freeze them in very small serves. So, it’s hard for me when I make the food, I know it’s good food and I’ve put protein in it and stuff, but she won’t eat it.” (P7)

#### 3.2.2. Theme 2: The Importance of a Healthy Diet

Eight carers acknowledged that a healthy diet is important in dementia care. Consuming fresh fruits and vegetables, fewer processed foods, reducing alcohol consumption, maintaining hydration, and lowering sugar and salt intake were common responses. Carers reported that a healthy diet may provide the nutrients and essential vitamins for brain health.

“It is important, yes because I think the brain will function better if the person eats healthy food. Like more broccoli, cauliflower, more spinach, green vegetables, lots of water. Yeah, and more nuts. So, I think the brain will function better if it’s healthy food rather than just pizza or pasta or canned food, for example.” (P6)

##### Sub-Theme 1: Adapting to the Person and Being Creative Are Essential

For six carers, understanding the feelings and food preferences of their loved one is crucial to providing a healthy diet. Making mealtimes an enjoyable experience, and providing multiple food options and presenting appealing food were part of the strategies used to improve food intake.

“You know, I think enjoyment is really one of the one of the few things that that we can get out of life, and this was making, maybe be as big a message as enjoying food. And you know, it’s important to keep that going, and to make sure that while technically, you know, a healthy diet is being provided. It also needs to be really tasty, and to make people want to eat rather than you know, getting through mechanically like you need fuel.”(P10)

##### Sub-Theme 2: Carers Aim to Provide a Healthy Dietary Pattern

Seven carers mentioned the benefits of a healthy dietary pattern, such as a Mediterranean-style diet. Carers learned of its benefits in nutrition education materials, including the Dementia Australia website. Whilst believing in a healthy dietary pattern to improve overall health, a carer reported that nutrition for people with dementia should not be different from any other person.

“I’ve been very, very keen to follow up research on topics such as intermittent fasting, you know, low carb, low salt diets, etc. [They’ve] come from a variety of sources, many of them online, but usually from reliable sources, fully qualified people who have done research and are open about what works and what doesn’t. The Mediterranean diet seems to be the winner at the moment, and I think that would go for the general population, but also for people who have dementia. [..] I think you need to eat well [and] cut out as many processed foods as possible.” (P10)

##### Sub-Theme 3: Involving the Person with Dementia and Social Eating Can Improve Nutrition

Five carers emphasised involving the person they care for in food preparation and social eating as important parts of life. A carer noted that sitting at a table and chatting while eating meals improves food intake.

“Well, we couldn’t use the table at the same time. And so, I found that [if] [person with dementia] was sitting watching TV, […], we had to stop that quickly, because that isolates him. Yeah, so we sit, and we talk, and he always says ‘oh, this is a great meal’. I mean he doesn’t always know where his meals come from. But presented nicely, you know, he’s happy and eats it all.”(P11)

Two carers recognised that engaging the person with dementia in meal preparation strengthens the relationship, giving them a sense of accomplishment and improving their quality of life.

“I think the social contact helps him a lot. He likes when someone is with him to eat together. Yeah, and it helps him a lot when we cook when I pass things. So, I allow him, I allow him sometimes to cook or let’s cook together! Yeah, that helps a lot, when the social connection and interaction, when we cook together.”(P6)

“There’s one meal he makes which is pasta, because we keep [arti]chokes, we can make, we make fresh pasta with our own eggs. But with some assistance, he can still do that. So, he gets a huge sense of achievement from doing that.”(P4)

##### Sub-Theme 4: The Association between Diet, Lack of Exercise, and Sedentary Behaviour

Five carers attribute the loss of interest in food to the increased sedentary behaviour and lack of exercise noticed in people with dementia. The presence of comorbid conditions also impacted this.

“[T]hey don’t do much exercise, you know, if you don’t move, you’re not going to develop an appetite.”(P1)

“His physical condition [heart issues] doesn’t let him to be active. So, I think that’s why he’s lost weight”(P11)

On the contrary, one carer found that the reduced physical activity in the person she cared for had shown the opposite effect, developing an insatiable appetite.

“He’s more sedentary these days, so he’s not as active. So, there’s a tendency to overeat.” (P4)

#### 3.2.3. Theme 3: Carers Seek Accessible and Clear Information on Nutrition

Nine carers had previously accessed education materials on nutritional care and its importance in dementia onset and progression. Carers reported having pursued this knowledge online, using the Dementia Australia website and other official websites. Four carers completed the University of Tasmania Understanding Dementia—Massive Open Online Course (UD-MOOC).

“I have done the University of Tasmania UD-MOOC online for dementia. So, I understand that one of the risk factors can be diet. I’m aware of a healthy diet being beneficial to anybody during their life.” (P7)

Carers sought advice on preventing dementia onset as they believe they were at a higher risk of developing dementia themselves and were interested in information on dietary guidelines and the connection between dementia and nutrition.

“I’m right behind her that I guarantee you this will be me and in 20 or 30 years. I know that I need to be thinking about my own brain health.” (P7)

Carers felt confused regarding the amount of unreliable or contradictory information found on the internet and wanted more accessible information, with less medical jargon informing sensible choices for the person they cared for.

“If you search on the internet for information about healthy eating, there is a plethora of information, often inconsistent and even contradictory. It isn’t easy for people to decide which sources of information to rely on.” (P4)

For carers, being handed easy-to-read pamphlets or booklets on nutrition education during medical consultations might assist them in addressing nutrition-related issues.

“I don’t think enough information is given to people from the doctors. […] I think that even if the doctors had a handout. That’s easy for a doctor to hand out and give to the person and say: Do you understand this? If you’ve got any problems…?” (P11)

“A factsheet or something, that would help. These are the foods to eat, these are the foods to avoid, these really help, these don’t help.” (P5)

Carers requested the information to be up-to-date, evidence-based, and directed more towards them and their needs, enabling them to make the healthiest nutrition choices.

“I’d hope to learn what science has, not just one study, you know, really solid peer base review research as to what’s coming out as the best way or is there [a] definite solid link with some foods perhaps to avoid and other foods to gravitate towards to have more.” (P7)

“I’d like to learn, I guess, information that reinforces what you know, and is up to date, I mean, things that [are] changing, information and ideas, and guidelines change over time.” (P1)

#### 3.2.4. Theme 4: Inadequate Support from Healthcare Professionals

Eight carers felt they were not receiving the support needed from healthcare professionals regarding nutrition due to insufficient training on nutrition-related issues or lack of interest.

“GP never talked about nutrition. Never. So, I also don’t think it’s her area of interest or expertise.” (P3)

“If I can be really honest, I don’t think they understand dementia themselves [GPs], and I don’t think they care.” (P5)

According to carers, the disease is often diagnosed at a later stage, resulting in feeling left alone, unable to make a difference. They were unsatisfied with the help their GPs or geriatricians provided on nutrition. Carers also mentioned the relatively high turnover of medical doctors and that they were not being referred to specialists.

“We have changed doctors, […] we weren’t very satisfied with how much help we were getting. But we were desperate for any advice and education, and he kind of went ‘ah you know, you can go online there, you know, you can Google’ which didn’t really help us. But he’s never focused very much on diet beyond general health, or weight loss, and suggesting dieting for weight loss, but has never really provided any advice around her cognitive decline.”(P8)

Commenting on GPs’ knowledge of dementia care, a carer believed that delegating the task of nutrition education to health professionals specifically trained in dementia care could improve care quality.

“It’s all about the other services that the GPs need to know about. And we’re saying it all the time. Their knowledge isn’t enough, so what we need is like dementia-trained nurses in a practice or something like that, and they would be the ones that would feed all this information regarding nutrition.” (P11)

Another carer said a 15-min consultation is not enough to cover all the issues they faced and that their GP focused on issues that they thought were more urgent.

“I mean, I think you know, in a 15-min discussion, there’s often lots of other things going on. And I think they just probably don’t even think about it. Diabetics, yeah, and other things, probably not.” (P1)

##### Sub-Theme 1: GPs Do Not Offer Referral to a Dietitian

Eight carers were not referred to see a dietitian. One carer emphasised that a referral to a dietitian should be offered to all people with dementia, irrespective of whether they are experiencing a nutrition-related issue.

“I know that when I’ve worked, I’ve worked in aged care. So, we have used dietitians in aged care for people with problems, they should be used for everybody, not just for people with problems.” (P11)

A carer reported that she would have been very interested in meeting a dietitian to raise issues concerning her partner. However, the GP did not refer her.

“I think it would be a good idea! I don’t think the importance of diet relating to dementia is widely known, and I would be interested in seeing a nutritionist or a dietitian in respect [to] my partner, yes. But he hasn’t done that.” (P4)

## 4. Discussion

The findings of this study suggest that informal carers of people with dementia possess basic nutrition knowledge. While most carers reported at least one type of nutrition-related issue, they had not been referred to a dietitian by their GP. Carers were aware of the benefits of a healthy dietary pattern and its association with cognitive function but found it challenging to provide this to the person they cared for and were unsatisfied with the support, help, and advice received to improve the nutrition of a person with dementia. Carers also wanted clear nutrition education materials adapted to their level of knowledge. This study is one of the first to assess the nutrition knowledge of informal carers of people with dementia and explore their experiences of acquiring nutrition knowledge. Similarities between the quantitative and qualitative data were identified, with carers reporting limited use and availability of dementia-specific nutrition resources in both the survey and interviews. Few participants reported having consulted a dietitian in the survey, and this was confirmed in the interviews, although carers wanted more nutritional advice specific to a person with dementia. Examples of nutrition-related issues reported in the interviews were consistent with experiences described in the interviews, particularly around the presence of multiple issues impacting nutrition, lack of appetite, and forgetting to eat.

Globally, a lack of nutrition knowledge among carers has been identified [19]. While we found no significant difference in the perception of providing a healthy diet to people with dementia between genders, female carers had greater nutrition knowledge than their male counterparts, suggesting a greater need for nutrition education amongst males. Previous research has demonstrated an association between nutrition knowledge, age, and level of education [35], and female carers in our sample had higher levels of education and interest in nutrition. In addition, historically and culturally, women traditionally provide food in the household, potentially explaining the overall higher nutrition knowledge [36]. Higher scores in the dietary guidelines and knowledge of food choices sections may have resulted from government campaigns [26,27,37]. However, lower knowledge in the food choices and the diet–disease relationship sections suggests that carers experienced difficulty in understanding more complex nutrition information compared with basic guidelines and general information [26,27,38]. A similar study using the validated GNKQ, which compared scores of carers of people with an intellectual disability with scores from the Australian community, concluded that carers lacked nutrition knowledge, as their scores were lower (56.6%) compared with the broader community (67.2%) [39].

The most common issue described by informal carers was the lack of support from healthcare professionals and referrals to a dietitian, which is consistent with previous research suggesting that carers feel unsupported and uninformed about nutritional care [12]. In this study, carers had wanted to see a dietitian, even if the person they cared for was not experiencing nutrition-related issues. This is consistent with findings that carers have confidence in a dietitian’s ability to answer nutrition-related questions [12]. In the earlier stages of dementia, advice from healthcare professionals may enable carers to learn how to ameliorate mealtime issues and develop effective coping strategies [23]. Support for decision making regarding nutrition and hydration for people with dementia should be a collaborative process, particularly for carers or family members of people with severe dementia [21,40].

Our findings suggest that carers are seeking additional nutritional information, more support from healthcare professionals regarding nutrition, and reliable and practical resources that address their individual circumstances and needs. Self-managed online resources were the preferred method of delivery by carers, and this modality offers flexibility and convenience in learning, and ease of access [41]. Carers also wanted resources that provide practical advice adapted to their level of knowledge. Previously, nutrition education resources have been found to be overwhelming and not matching carers’ expertise and experiences [12]. Similarly, carers expressed a preference for succinct information, less jargon, and more information on food-related processes in studies from Singapore [42], Canada [43], and the United Kingdom [44]. In our study, carers described nutrition resources as confusing or contradictory at times. A review of online nutrition resources for the prevention and treatment of dementia were found to be of good overall quality, but information on eating issues was lacking [45]. Nutrition education programmes may enable carers to develop cooking skills, interpret food labelling, and detect nutrition-related issues in people with dementia [46,47]. However, to meet carers’ needs, stakeholder engagement and co-design during nutrition education programmes are critical [47].

Several strategies have been identified that may support carers in improving nutritional intake. This includes nutrition screening and assessment, training and education programmes, mealtime environment and routine modification, nutritional supplements, and artificial nutrition and hydration during the later stages of dementia [16]. Non-professional support strategies such as domestic help or lunch clubs are helpful for carers who find food preparation difficult [48]. In medical centres, professional support could be provided by primary care nurses who offer person-centred health care [49]. Practitioners have also acknowledged insufficient training to provide nutrition advice to carers, particularly in the later stages of dementia [22,29], and concerns about quality dementia care and inadequate knowledge of dementia by healthcare professionals have been raised in China, the United States, and Europe [50,51,52]. Furthermore, Nair et al. [53] found that culturally diverse informal carers in the United Kingdom experienced difficulty in accessing appropriate nutritional care resources, prompting healthcare professionals to improve cultural awareness and provide contextually relevant advice to carers.

In our survey, a third of carers believed that they did not provide a healthy diet. Despite its benefits, multiple barriers exist to providing a healthy dietary pattern, such as lack of family support, seasonal foods, time constraints, cooking skills, or work commitments [54]. Moreover, mealtime difficulties make healthy-diet provision challenging. Depending on the person or the stage of dementia, various eating disorders can be observed, such as geriatric anorexia occurring in the middle or final stages of dementia, or hyperphagia, an abnormal increase in appetite [55]. Mealtime issues coupled with physiological changes due to the progression of dementia and its emotional impact often affect eating patterns and may lead to malnutrition [10]. However, mealtimes can allow the caregiving dyad to share food and encourage social interaction, improving food intake [56]. To optimise mealtime experiences, carers reported sitting together, socialising, attractively presenting food, and allowing enough time to eat as strategies to increase food intake and improve nutritional status [46]. Accommodating the taste and food preferences of the person they care for is another strategy for addressing malnutrition [12]. For the carers in our sample, involving the person with dementia in meal preparation gave a sense of achievement, improved intake, and contributed to improving the dyad’s quality of life. Nonetheless, developing coping strategies based on professional advice may further help carers improve mealtime experiences, identify nutritional risks, and prevent malnutrition [14].

Carers expressed a desire to learn about foods that may improve cognition or help prevent dementia onset. This interest by carers in different types of food and how foods may affect the brain and the rate of memory loss has been previously identified [48]. However, carers are aware of the difficulties involved in providing a nutritious diet and ensuring the person with dementia is enjoying their food [14] and may prefer to offer food that the care recipient is willing to eat, rather than particular foods that lower the risk of cognitive decline [57]. Several nutrient-dense foods may improve cardiovascular risk and delay the onset of Alzheimer’s disease [58]. Given that the literature indicates that no specific diet, supplements, or lifestyle interventions conclusively slow cognitive decline [59,60], a focus on overall nutritional intake is likely beneficial.

### Limitations

The survey sample size was relatively small and not generalisable to all carers. Carers of people with dementia often balance multiple responsibilities in addition to their caring duties; therefore, the length and limited available time to complete the survey may have contributed to the small sample size. In addition, despite the increasing use of technology by older people, this sample may be limited to those who felt comfortable accessing and completing an online survey. Most participants were from New South Wales (52%), and participants may not be representative of the Australian population. Due to the randomised all-female sample in the follow-up interviews, our conclusions from the qualitative data do not incorporate the views of male carers on nutrition-related issues or their needs regarding nutrition education programmes. The random sample used in the qualitative interviews represents a limitation, as a purposeful sample may have elicited more in-depth and rich data. Finally, the education level of the carers interviewed was relatively high, and some interviewees had a healthcare background. Higher education is associated with a higher socioeconomic status, which may affect dietary knowledge and the ability to provide a high-quality diet.

## 5. Conclusions

This exploratory study identified that informal carers of people with dementia had basic nutrition knowledge. Carers’ nutrition gaps were in more complex nutrition knowledge, identified by challenges in understanding the diet–disease relationship and the nutrient components in foods. Carers preferred self-managed online nutrition resources but also wanted a referral to access a dietitian. Carers also expressed a preference for clear nutrition education materials adapted to their level of knowledge, combined with better support from their health practitioners. Given that people with dementia have specific dietary needs that depend on disease stage and change with disease progression, nutrition knowledge programmes should be designed to address these needs. To improve nutritional status among people with dementia, nutrition programmes for carers should be co-designed and include practical information, while further dementia-specific training for healthcare professionals is needed. Access to greater nutrition knowledge and professional support could help address nutrition-related issues and reduce the risk of nutritional deficiencies and malnutrition in people with dementia, contributing to improving the quality of life of both people with dementia and their carers.

## Figures and Tables

**Figure 1 geriatrics-08-00077-f001:**
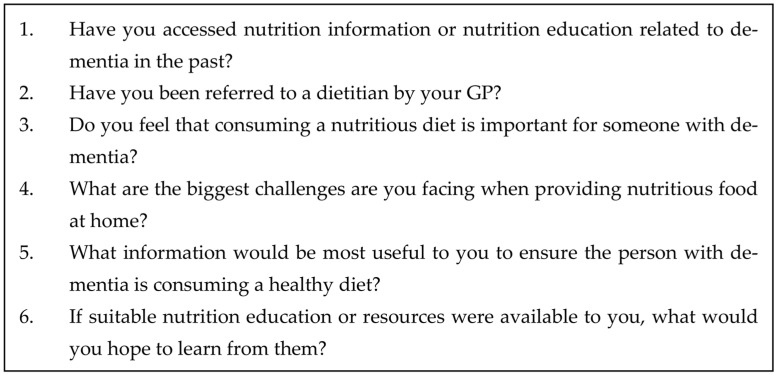
Semi-structured interview questions for informal carers.

**Figure 2 geriatrics-08-00077-f002:**
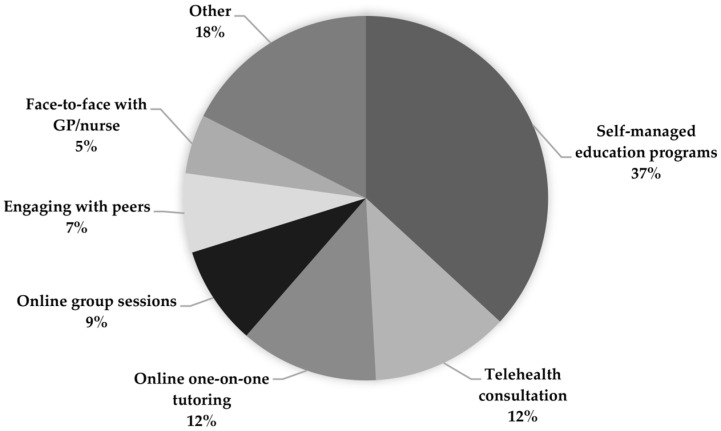
Informal carers’ preferred methods of nutrition education.

**Table 1 geriatrics-08-00077-t001:** Background and sociodemographic information on survey respondents.

	All Carers	Female Carers	Male Carers	*p*-Value
n	57	44	13	
	63.0 ± 13.1	61.5 ± 12.1	68.2 ± 15.3	0.108
Education:				0.475
Certificate/diploma	11	8	3	
Year 10 or equivalent	3	3	0	
Year 12 or equivalent	4	2	2	
University degree	24	18	6	
Higher university degree	15	13	2	
Time spent being a carer:				0.674
<6 months	2	2	0	
6 months–1 year	5	3	2	
1 year–2 years	6	5	1	
2 years to 5 years	23	19	4	
5 years to 10 years	21	15	6	
Relationship:				0.786
Spouse	26	18	8	
Child	16	13	3	
Grandchild	1	1	0	
Other relative	8	6	2	
Neighbour/friend	1	1	0	
Personal care worker	3	3	0	
Other	2	2	0	
Retirement:				0.022
Yes	28	18	10	
No	29	26	3	
Area of residence:				0.943
Urban	39	28	11	
Regional	18	16	2	
Ethnicity:				0.505
Australian	34	27	7	
African	1	1	0	
British	8	7	1	
Chinese	3	1	2	
European	6	5	1	
South Asian	2	1	1	
Other	3	2	1	

**Table 2 geriatrics-08-00077-t002:** AUS-R NKQ scores and questionnaire results.

	All Carers	Female Carers	Male Carers	*p*-Value
n	57	44	13	
AUS-R GNKQ scores	22.9 ± 4.57	23.8 ± 4.51	20.0 ± 3.53	0.007
Resources accessed by carers:				0.213
Consulted a dietitian	8	7	1	
Consulted GP	6	4	2	
Specialist (e.g., geriatrician)	5	4	1	
Advice from friend/family	2	2	0	
Read a book about nutrition	3	3	0	
Course on nutrition	1	0	1	
Internet	1	0	1	
Multidisciplinary team	15	12	3	
Did not answer	16	12	4	
Perceived balanced diet provided by the carer:				0.754
Yes	36	28	8	
No	13	9	4	
Not sure	6	5	1	
Change in protein consumption:				0.884
Not sure	2	2	0	
Increase	5	3	1	
Decrease	29	22	7	
No change	21	16	5	
Sweet foods craving:				0.139
Not sure	2	2	0	
Increase	27	19	8	
Decrease	12	12	0	
No change	16	11	5	
Fruit and vegetable intake:				0.855
Not sure	2	2	0	
Increase	10	8	2	
Decrease	22	17	5	
No change	23	17	6	
Consumption of salty foods:				0.755
Not sure	2	2	0	
Increase	7	6	1	
Decrease	10	8	2	
No change	38	28	10	
Observed weight change:				0.655
Weight gain	10	7	3	
Weight loss	31	25	6	
No change	12	8	4	
Weight fluctuation	1	1	0	
Did not answer	3	3	0	
Nutrition-related issues:				0.900
No issues	12	8	4	
Dysphagia	2	2	0	
Forgetting to eat	5	4	1	
Chewing difficulties	2	2	0	
Functional difficulties	3	2	1	
Refusing offered food	10	8	2	
Combination of multiple issues	23	18	5	

**Table 3 geriatrics-08-00077-t003:** Sociodemographic characteristics of carers who participated in interviews.

Code	Age	Ethnicity	Dementia Type	Relationship	Years as a Carer	Place of Residence	Occupation	AUS-R NKQ Result
P1	69	Australian	Unsure	Family member	5–10	NSW	Nurse	31
P2	61	European	Vascular	Live-in carer	5–10	QLD	Live-in carer	24
P3	64	British	Lewy Body	Child	5–10	NSW	Psychologist	21
P4	69	European	Alzheimer’s	Spouse	2–5	SA	Retired	24
P5	56	Australian	Alzheimer’s	Child	2–5	NSW	Social worker	18
P6	51	European	Alcohol-related	Spouse	5–10	NSW	Social worker	21
P7	68	Australian	Vascular	Family member	2–5	SA	Retired	26
P8	53	Australian	Alzheimer’s	Child	1–2	WA	Writer	24
P9	78	Australian	Alzheimer’s	Spouse	2–5	NSW	Retired	24
P10	67	Australian	Alzheimer’s	Spouse	2–5	NSW	Retired	29
P11	77	Australian	Alzheimer’s	Spouse	5–10	NSW	Retired	22

**Table 4 geriatrics-08-00077-t004:** Themes and sub-themes.

Theme	Sub-Theme
Providing a healthy diet is challenging for carers	1.1 Physiological changes make it difficult to provide a healthy diet 1.2 Memory loss contributes to lower energy intake 1.3 Financial barriers to healthy nutrition 1.4 The provision of a healthy diet accentuates carer burden
2. Carers recognise the importance of a healthy diet	2.1 Adapting to the person and being creative are essential 2.2 Carers aim to provide a healthy dietary pattern 2.3 Involving the person with dementia and social eating can improve nutrition 2.4 The association between diet, lack of exercise, and sedentary behaviour
3. Carers seek accessible and clear information on nutrition	
4. Inadequate support from healthcare professionals	4.1 GPs do not offer dietetic referral

## Data Availability

De-identified quantitative data are available upon request.

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
