# Peer review of "An Exploratory Study of Nutrition Knowledge and Challenges Faced by Informal Carers of Community-Dwelling People with Dementia: Online Survey and Thematic Analysis"

_geriatrics, 2023, doi:10.3390/geriatrics8040077_

Round 1
Reviewer 1 Report
Thank you for this interesting manuscript. It explored the knowledge and experiences of informal carers of people with dementia regarding nutrition. The study used mixed methods, which offered us some useful insights. The manuscript was quite well-written; however, it could be improved by adding and clarifying some points. I would encourage the authors to consider the following comments and revise the manuscript accordingly.
11. Introduction, paragraphs 1 and 2: The authors may wish to reduce and combine paragraphs 1 and 2 for a general description of dementia and then focus on eating problems in dementia. For example, the authors have not yet mentioned the cause and progression of eating disorders and how they could have negative impacts on people with dementia and their family carers to demonstrate the gaps and significance of the study. In dementia, it is also more than only options of nutritious diets, but also other kinds of support and encouragement, according to eating problems and needs of people with dementia. This should be briefly included in the introduction.
22. Introduction, paragraph 4: Knowledge about dementia and related eating disorders influences and is necessary for shared decision-making for nutritional interventions, especially in the later stages where family carers need to make decisions for people with dementia who have limited decisional capacity. The authors may wish to include a recent study on the decision-making process about nutrition and hydration for people with dementia, which highlights the importance of knowledge and information exchange in the decision-making process (please see Anantapong et al. 2023, https://doi.org/10.1002/gps.5884)
33. Data collection, 2.3.2 Qualitative data, using a random number generator: Selection bias is not relevant as in qualitative research, so it might not be required. The purposive sample would be more justified for the qualitative part to ensure that participants would have direct experiences and give in-depth, rich data. Some revisions would be needed to follow related theoretical assumptions of qualitative research.
44. Data analysis: The authors should check Braun and Clarke (2020), https://doi.org/10.1080/14780887.2020.1769238) and reconsider if the authors actually used/followed the procedures of reflexive thematic analysis (TA) or other types of TA and revise the methods section accordingly. In many previous studies, there have been some common misunderstandings and incompatibilities, and this reflects in the methods section of the current study. For example, theme consensus is normally not required or even considered counterproductive in reflexive thematic analysis.
55. Results, Page 6, ‘… participants had more difficulty with the food groups and their nutrients …’ What are the food groups? Need revision for accuracy.
66. Discussion: Overall, these were quite interesting and well-written discussions. However, as the use of mixed methods is the strength of this study, the authors may wish to add some critical discussion to explicitly link between the quantitative and qualitative findings. For example, how the low levels of knowledge (or sub-scales) among the participants could be explained by findings from qualitative data (themes/quotes). There are quite limited connections between the two datasets. Think about how the qualitative data have helped the authors (and readers) understand or interpret quantitative data (and vice versa).
Author Response
Comments and Suggestions for Authors
Thank you for this interesting manuscript. It explored the knowledge and experiences of informal carers of people with dementia regarding nutrition. The study used mixed methods, which offered us some useful insights. The manuscript was quite well-written; however, it could be improved by adding and clarifying some points. I would encourage the authors to consider the following comments and revise the manuscript accordingly.
Thank you for your kind comments. We have provided an itemised response to your suggestions below.
- Introduction, paragraphs 1 and 2: The authors may wish to reduce and combine paragraphs 1 and 2 for a general description of dementia and then focus on eating problems in dementia. For example, the authors have not yet mentioned the cause and progression of eating disorders and how they could have negative impacts on people with dementia and their family carers to demonstrate the gaps and significance of the study. In dementia, it is also more than only options of nutritious diets, but also other kinds of support and encouragement, according to eating problems and needs of people with dementia. This should be briefly included in the introduction.
The authors would like to thank you for this suggestion. We have now revised the introduction as suggested and included information about the cause and progression of eating difficulties in people with dementia in the second half of the second paragraph.
- Introduction, paragraph 4: Knowledge about dementia and related eating disorders influences and is necessary for shared decision-making for nutritional interventions, especially in the later stages where family carers need to make decisions for people with dementia who have limited decisional capacity. The authors may wish to include a recent study on the decision-making process about nutrition and hydration for people with dementia, which highlights the importance of knowledge and information exchange in the decision-making process (please see Anantapong et al. 2023, https://doi.org/10.1002/gps.5884)
Thank you for this important point. We have now included this information and cited the recommended article on Page 2, Line 74-86. We have also identified another relevant article by these authors which we have incorporated into the discussion on Page 14, Line 520.
- Data collection, 2.3.2 Qualitative data, using a random number generator: Selection bias is not relevant as in qualitative research, so it might not be required. The purposive sample would be more justified for the qualitative part to ensure that participants would have direct experiences and give in-depth, rich data. Some revisions would be needed to follow related theoretical assumptions of qualitative research.
Thank you for your comments. We agree a purposeful sample would have been suitable and could have provided more in-depth and rich data. In our study, given all participants were required to complete the online survey in the first instance, we did not want to give an impression that they would then also be required to complete an online interview as well. Given carers of people with dementia are very busy and often under strain, we felt the best approach to increase the likelihood of survey responses was to inform participants they would be randomly invited to complete a follow-up interview. We have now included this information on Page 4, Line 152-153.
To clarify, we have added text in section 2.3.2 on Page 4, Line 152-15. and also acknowledged the limitations of this methods on Page 15, Line 577-578.
- Data analysis: The authors should check Braun and Clarke (2020), https://doi.org/10.1080/14780887.2020.1769238) and reconsider if the authors actually used/followed the procedures of reflexive thematic analysis (TA) or other types of TA and revise the methods section accordingly. In many previous studies, there have been some common misunderstandings and incompatibilities, and this reflects in the methods section of the current study. For example, theme consensus is normally not required or even considered counterproductive in reflexive thematic analysis.
Thank you for this perceptive comment. We acknowledge that our application of TA requires clarification and we have done that in the data analysis section. We did use the 6-step process described by Braun and Clarke (2016) and we are happy that this is a contemporary way of qualitative analysis as this has been repeated by Braun and Clark (2021) in the citation you provide. What we have changed is the description of ‘consensus’ as on reflection this was not what we did and specifically described this as an inductive process as (for example) our coding was open and organic rather than using a pre-defined coding structure or framework. We have also added a section on reflexivity and positionality that we hope strengthens this section. Please section 2.4.
- Results, Page 6, ‘… participants had more difficulty with the food groups and their nutrients …’ What are the food groups? Need revision for accuracy.
Thank you for this suggestion. For clarity, we have now included a more detailed description of the survey in the Methods in Section 2.3.1 Quantitative data where we describe the data collected. This is to provide clarity on the results presented on Page 6.
- Discussion: Overall, these were quite interesting and well-written discussions. However, as the use of mixed methods is the strength of this study, the authors may wish to add some critical discussion to explicitly link between the quantitative and qualitative findings. For example, how the low levels of knowledge (or sub-scales) among the participants could be explained by findings from qualitative data (themes/quotes). There are quite limited connections between the two datasets. Think about how the qualitative data have helped the authors (and readers) understand or interpret quantitative data (and vice versa).
Thank you for this suggestion. We have now highlighted some of the connections between the survey data and interviews in the first paragraph of the discussion.
Reviewer 2 Report
I have concerned about the sample size, seems so small. Is it possible to present some of the results in figures?
English is good, some minor polishing is required
Author Response
Comments and Suggestions for Authors
I have concerned about the sample size, seems so small. Is it possible to present some of the results in figures?
The authors would like to thank Reviewer Two for the time take to read our work and provide suggestions for improvement. We agree, the sample size for the survey is small. However, this is a vulnerable and hard to reach population. We have acknowledged the small sample size in the limitations section. We have also added additional explanation at the start of section 2.4 at Page 3, Lie 162-165.
Comments on the Quality of English Language
English is good, some minor polishing is required
Thank you. We have thoroughly checked the writing in the manuscript and made appropriate changes.
Reviewer 3 Report
Thank you for giving me the time to review your manuscript. This manuscript is interesting and scientifically meaningful for considering nutrition knowledge and experiences of informal carers of community-dwelling people with dementia. Regarding the contents, the following revision should be considered.
The title should contain the significance of this study.
The abstract should contain the meaningfulness of this study.
Generally, there is no paragraph writing. The background contains many paragraphs. The author should focus on theory building, the problems, and research question paragraphs. The first and second paragraphs should include general information on nutrition knowledge and experiences of informal carers of community-dwelling people with dementia. Moreover, the third and fourth paragraphs should introduce the research question as the theoretical and conceptual framework, including nutrition knowledge and experiences of informal carers of community-dwelling people with dementia the authors focused on in international contexts and research questions.
The background should delineate the international contexts of nutrition knowledge and experiences of informal carers of community-dwelling people with dementia.
In the introduction, the researchers should show the research question clearly.
This study should describe why this study used mixed method approaches to investigate the research question clearly.
The sample calculation should be described in the analysis part.
The qualitative analysis was described insufficiently. What kinds of thematic analysis were used? And how did the authors integrate the qualitative and quantitative results? These issues are critical for the quality of this research.
The authors should precisely describe the reflexivity of the interviewers and their backgrounds for the research trustfulness.
In the discussion, the first paragraph should summarize the results.
The discussion should describe the research findings in international contexts. As the same as the background, this is a critical point for the publication of international journals.
Overall, the article is too long, not respecting paragraph writing. The authors should rewrite the manuscript comprehensively and focus only on the study’s research questions.
Author Response
Comments and Suggestions for Authors
Thank you for giving me the time to review your manuscript. This manuscript is interesting and scientifically meaningful for considering nutrition knowledge and experiences of informal carers of community-dwelling people with dementia. Regarding the contents, the following revision should be considered.
The authors would like to thank Reviewer Three for the time taken to provide feedback to improve our work.
The title should contain the significance of this study.
Thank you for this suggestion. We have updated the title to:
“An exploratory study of nutrition knowledge and challenges faced by informal carers of community-dwelling people with dementia: online survey and thematic analysis”
The abstract should contain the meaningfulness of this study.
Thank you for your suggestion. We have edited the abstract to more clearly convey the meaningfulness of the study.
Generally, there is no paragraph writing. The background contains many paragraphs. The author should focus on theory building, the problems, and research question paragraphs. The first and second paragraphs should include general information on nutrition knowledge and experiences of informal carers of community-dwelling people with dementia. Moreover, the third and fourth paragraphs should introduce the research question as the theoretical and conceptual framework, including nutrition knowledge and experiences of informal carers of community-dwelling people with dementia the authors focused on in international contexts and research questions.
The background should delineate the international contexts of nutrition knowledge and experiences of informal carers of community-dwelling people with dementia.
We agree with these valuable recommendations and have subsequently re-organised the introduction. In removing some of the specific focus on the Australian context, we have added information from an international perspective.
In the introduction, the researchers should show the research question clearly.
The research questions are now shown on Page 2, Line 90.
This study should describe why this study used mixed method approaches to investigate the research question clearly.
Thank you for this suggestion. We have now updated section 2.1 Study design to provide more clarity on the mixed methods approach. Please see Page 2, Line 102-113.
The sample calculation should be described in the analysis part.
Thank you for this suggestion. The sample size required for the results to be generalisable can now be found on Page 4, Lines 162-165.
The qualitative analysis was described insufficiently. What kinds of thematic analysis were used? And how did the authors integrate the qualitative and quantitative results? These issues are critical for the quality of this research.
We conducted an inductive thematic analysis guided by Braun and Clarke (2016) six step process. We acknowledge that this process and interpretation of it has developed since 2016, but we are encouraged that this process is still considered relevant (Braun and Clarke 2021). We do acknowledge that our description of the qualitative data analysis required clarification and we have done that in the revised paper. Please see Page 4 and 5, Lines 173-191.
The authors should precisely describe the reflexivity of the interviewers and their backgrounds for the research trustfulness.
The authors appreciate this suggestion and we have now included this information. Please see Page 4 and 5, Lines 173-191.
In the discussion, the first paragraph should summarize the results.
The first paragraph of the discussion is now updated to summarise the overall results, including description of some of the similarities in findings between the survey and interviews, as suggested by Reviewer 1.
The discussion should describe the research findings in international contexts. As the same as the background, this is a critical point for the publication of international journals.
We thank the Reviewer for this suggestion and have now included international context in the discussion. Please see Page 14, Lines 514-516 and Lines 529-536.
Overall, the article is too long, not respecting paragraph writing. The authors should rewrite the manuscript comprehensively and focus only on the study’s research questions.
Thank you for your comments. We have taken time to review the manuscript and writing. While we acknowledge the article is long, there is a large qualitative component to the article which is a major contributing factor. Indeed, Braun and Clarke have advocated for journals to increase their word limits to account for longer articles with qualitative components. Nevertheless, our introduction has been shortened and we have carefully edited the discussion to reduce the number of words. However, some parts have been added throughout the manuscript to address reviewer comments.
Round 2
Reviewer 1 Report
Thank you for the revision. The authors have successfully responded to the comments and revised the manuscript accordingly. I have no further comments.
Reviewer 3 Report
This manuscript has been revised to the quality of the journal.